# Value of Antibody Determinations in Chronic Dysimmune Neuropathies

**DOI:** 10.3390/brainsci13010037

**Published:** 2022-12-23

**Authors:** Stefano Tozza, Emanuele Spina, Aniello Iovino, Rosa Iodice, Raffaele Dubbioso, Lucia Ruggiero, Maria Nolano, Fiore Manganelli

**Affiliations:** 1Department of Neurosciences, Reproductive Sciences and Odontostomatology, University Federico II of Naples, 80131 Naples, Italy; 2Neurology Department, Skin Biopsy Laboratory, Istituti Clinici Scientifici Maugeri IRCCS, 82037 Telese Terme, Italy

**Keywords:** chronic dysimmune neuropathy, anti-MAG neuropathy, multifocal motor neuropathy with conduction block, autoimmune paranodopathies

## Abstract

Chronic dysimmune neuropathies encompass a group of neuropathies that share immune-mediated pathomechanism. Chronic dysimmune antibody-related neuropathies include anti-MAG neuropathy, multifocal motor neuropathy, and neuropathies related to immune attack against paranodal antigens. Such neuropathies exhibit distinguishing pathomechanism, clinical and response to therapy features with respect to chronic inflammatory demyelinating polyradiculoneuropathy and its variants, which represent the most frequent form of chronic dysimmune neuropathy. This narrative review provides an overview of pathomechanism; clinical, electrophysiological, and biochemical features; and treatment response of the antibody-mediated neuropathies, aiming to establish when and why to look for antibodies in chronic dysimmune neuropathies.

## 1. Introduction

Chronic dysimmune neuropathies encompass a group of neuropathies that share immune-mediated pathomechanisms.

In this context, the recognition of specific antibodies against antigenic targets of the peripheral nervous system (PNS) allows for identifying distinctive patients with respect to those affected by the most frequent form of chronic dysimmune neuropathy, namely, chronic inflammatory demyelinating polyradiculoneuropathy (CIDP) and its variants.

Accordingly, early recognition of antibodies with appropriate diagnostic testing is crucial for properly diagnosing and managing patients and starting potentially effective therapies.

The spectrum of CIDP includes CIDP and its clinical variants, including distal, multifocal, motor, and sensory subtypes [1]. The demyelinating features in a nerve conduction study (NCS) associated with inflammation, macrophage-mediated demyelination, and response to immune therapy (IVIg and steroids) represent the hallmarks of CIDP and its variants. 

On the other hand, chronic dysimmune antibody-related neuropathies include anti-

MAG neuropathy, multifocal motor neuropathy, and neuropathies related to immune attack against paranodal antigens. Such neuropathies exhibit distinguishing pathomechanism and clinical and pathological response to treatment features with respect to CIDP [1].

This narrative review provides an overview of pathomechanism; clinical, electrophysiological, and biochemical features; and treatment response of antibody-mediated neuropathies, aiming to establish when and why to look for antibodies in chronic dysimmune neuropathies.

## 2. Anti-MAG (Myelin-Associated Glycoprotein) Neuropathy

Anti-myelin-associated glycoprotein (MAG) neuropathy is a rare disease with a prevalence of approximately 1 in 100,000 [2,3], although representing the most common paraproteinemic neuropathy, which predominantly affects elderly males [4]. Anti-MAG neuropathy is a demyelinating neuropathy mediated by a monoclonal IgM antibody that binds to MAG protein and, different from CIDP, is not considered an inflammatory disease, and therefore, typical CIDP treatments are usually only transiently effective in these patients. EAN/PNS guidelines suggest searching for monoclonal gammopathy in all patients with suspected CIDP [1], and if an IgM paraprotein is present, anti-MAG antibodies must be tested.

### 2.1. Pathomechanism

MAG is a minor constituent of peripheral nervous system myelin, and it is located in the periaxonal space (the innermost myelin membrane wrap) between myelinating cells and axons [5], but also in paranodal loops as well as Schmidt-Lanterman incisures, and thus, it is exposed to the extracellular space and easily accessible to autoantibodies [6]. 

MAG is involved in signalization between Schwann cells and axons, in enhancing long-term axon–myelin stability and attachment [7,8], and in neurofilament phosphorylation signaling, resulting in axonal caliber increasing [9]. Moreover, MAG participates in defining the distribution of axon molecules at nodes of Ranvier [10,11,12]. 

Interestingly, the reactive determinant of an anti-MAG antibody is due to the complex sulfated trisaccharide on MAG known as the HNK-1 epitope [13,14]. In the early stage, anti-MAG antibodies bind to the paranodal region, impairing saltatory conduction [15] and inducing nodal and paranodal molecular alterations [16]. Subsequently, morphological changes occur with a loss of myelinated fibers, reduced axonal caliber [9], thinned myelin sheaths, and widely spaced myelin lamellae in the outermost myelin lamellae [17,18,19,20]. Moreover, most studies confirm the presence of complement components, such as C3d or C5 [21,22,23], suggesting that these complement components may be effectors of detachment of terminal loops from the axolemma at the node of Ranvier, resulting in myelin widening in the internode. This detachment may be the primary cause of axonal damage and subsequent axonal loss in anti-MAG neuropathy [16,24]. 

### 2.2. Clinical Features

The typical phenotype is characterized by chronic and insidious predominantly sensory polyneuropathy, usually affecting initially the lower extremities with paraesthesia and dysesthesia as well as cramps [25]. Sensory ataxia with gait imbalance is frequently observed, and some patients develop intentional tremor in the upper limbs [26]. 

Although some patients may complain of worsening over some months or a few years, disease is generally slowly progressive over decades, developing significant disability over time [27] with muscle weakness and wasting, severe ataxia, and intention tremor with functional impairment of the hands. A higher age at onset is associated with a higher risk of disability [28,29].

Nevertheless, about 20% of patients present an atypical clinical phenotype characterized by a precocious weakness of proximal muscles resembling the typical presentation of CIDP (CIDP-MAG) [26,30,31,32]. The management of these patients is challenging as it is still unclear whether they are part of the anti-MAG neuropathy spectrum or represent a CIDP with false-positive anti-MAG.

### 2.3. Electrophysiological Features

Electrophysiological testing represents the first-line investigation in the diagnosis of anti-MAG neuropathy. The typical electrophysiological pattern is a demyelinating neuropathy characterized by disproportionate distal slowing (prolonged distal motor latency with mild reduction of motor nerve conduction velocity) [33] and slower conduction velocity in an entrapment site (e.g., ulnar nerve at the elbow) [34]. Moreover, a mild axon loss can occur in the early stage of disease as well [35,36]. The peculiar demyelination distribution presumably is related to the anatomically leakier blood–nerve barrier at the distal nerve terminals, where the IgM antibody with great molecular weight can easily be accessed and thus binds neuronal structures [37]. A terminal latency index (TLI) of 0.26 in the median nerve and 0.33 in the ulnar nerve represents the threshold values for distinguishing anti-MAG neuropathy from CIDP-MAG [38]. Other parameters, such as a modified F-ratio or residual latency, can help to distinguish anti-MAG neuropathy from CIDP [39]. 

### 2.4. Biochemical Features

The diagnosis of anti-MAG neuropathy is based on detecting the presence of anti-MAG antibodies. Anti-MAG antibodies can be detected by ELISA, a more sensitive and reliable screening method for determining anti-MAG antibodies than Western blotting [26]. However, there is still debate concerning the ideal cut-off value for the positivity of anti-MAG autoantibodies from ELISA Buhlmann Diagnostics [26,40]. Although the manufacturer established a cut-off of 1.000 BTU (Buhlmann titer units), the best clinical response to treatment with rituximab was correlated with an anti-MAG titer ≥ 10.000 BTU [26], and more recently, one study established the cut-off with the best combination of sensitivity and specificity at a threshold of 7.000 BTU [40]. Nevertheless, false-positive and false-negative cases still represent a main problem in clinical practice, and it is therefore important to perform a careful clinical and electrophysiological assessment of patients with suspected anti-MAG neuropathy. 

As the antigenic epitope of MAG resides in the HNK-1 epitope [7,41,42], a new ELISA using this carbohydrate has been developed. The anti-HNK1 ELISA has high sensitivity (98%) and specificity (99%) in the diagnosis of anti-MAG neuropathy, and anti-HNK1 titers are correlated to the disease severity, suggesting that this test could be used as an outcome measure in clinical trials [43]. 

Since anti-MAG antibodies are found in more than 70% of patients with typical M component-related demyelinating polyneuropathy [4], in clinical practice, an anti-MAG antibody test is performed only in patients with detectable IgM monoclonal gammopathy. However, some cases of anti-MAG neuropathy can lack IgM monoclonal gammopathy [44,45,46,47] with the disclosure of the IgM monoclonal protein after years of follow-up [45]. Although these reports are anecdotical, it is suggested to test anti-MAG antibodies in patients with distal chronic sensorimotor demyelinating neuropathy, regardless of the detection of IgM monoclonal gammopathy.

### 2.5. Radiological Features 

Interestingly, although the clinical and electrophysiological involvement is typically distal in anti-MAG neuropathy, brachial plexus MRI has shown nerve hypertrophy in about 40% of patients, indicating a more generalized involvement in this neuropathy. Moreover, altered diffusion in the nerve roots is inversely correlated with disease duration, suggesting a loss of myelin integrity due to the demyelination process [48]. 

Nerve ultrasound (US) studies confirm widespread nerve enlargement, not confined to the more distal parts of nerves, with the greatest enlargement at common entrapment sites [49,50,51]. However, the role of US in distinguishing anti-MAG neuropathy from CIDP remains debated. 

### 2.6. Differential Diagnosis

The distal CIDP variant is a clinical condition that exactly resembles anti-MAG neuropathy, presenting with distal sensory loss, gait instability, and distal weakness. Since about two-thirds of patients with this phenotype have IgM paraproteinemic neuropathy and often anti-MAG antibodies [52], EAN/PNS guidelines recommend testing an IgM protein and an anti-MAG antibody in patients with distal CIDP phenotype and repeating them if they are negative [1]. Over anti-MAG positivity, specific electrodiagnostic (disproportionately prolonged motor latencies) and morphological (demyelinating features without inflammation) findings associated with poor response to immunomodulatory therapies can help in differential diagnosis.

Anti-MAG neuropathy with atypical phenotype can be clinically indistinguishable from CIDP with M protein and positivity of anti-MAG (CIDP-MAG). However, the treatment totally differs between anti-MAG and CIDP. In fact, CIDP treatments (IVIg or steroids) are not or only transiently effective in anti-MAG patients. Moreover, in anti-MAG patients, hematological follow-up is required due to the progression risk of M-component to hematological malignancies, such as multiple myeloma, Waldenstrom macroglobulinemia, amyloid light-chain (AL) amyloidosis, with an approximate risk of 1% per year [53]. 

Recently, Doneddu and colleagues developed a diagnostic score for differentiating anti-MAG neuropathy from CIDP-MAG [54]. The score uses a clinical and electrophysiological characteristic to support or not the diagnosis of anti-MAG neuropathy. Though the score is not able to properly categorize CIDP-MAG in anti-MAG neuropathy and CIDP, it could predict the response to IVIg. In fact, CIDP-MAG patients with a score lower than 0 respond to IVIg in over 70% of cases. According to EFNS/PNS recommendation, the authors suggest considering IVIg treatment in patients with anti-MAG neuropathy clinically similar to typical CIDP [1].

Another differential diagnosis is hereditary neuropathy with liability to pressure palsy (HNPP). HNPP can share with anti-MAG neuropathy the clinical phenotype (distal sensory disturbance), electrophysiological findings (mild reduction of nerve conduction velocity slower in the entrapment site), and typical morphological aspect of tomacula (nerve swelling) in nerve biopsy [55]. However, HNPP can manifest a history of palsy and pes cavus and a positive family history, helping in distinguishing it from anti-MAG. Moreover, in anti-MAG neuropathy compared with HNPP in [34], absolute values of distal motor latencies and conduction velocities outside entrapment sites were slower and amplitudes potential were lower. 

### 2.7. Therapy

As anti-MAG neuropathy is the most common disabling paraproteinemic neuropathy and since the anti-MAG antibodies appear to exert a direct pathogenic effect on the myelin structure and function, B cells depleting therapies have been the main therapeutic mode of treatment. Initially, chemotherapy treatments (chlorambucil, cyclophosphamide, or fludarabine) were used, but due to their toxicity or risk of secondary malignancies, they have been replaced by anti-CD20 antibodies, such as rituximab [56]. However, different studies show that rituximab improves the clinical condition only in 30–50% of anti-MAG patients [26,57]. The low response may be in part due to the mild clinical picture (distal sensory disturbances), which can make it difficult to see real improvement. For this reason, in Italy, rituximab was approved only for anti-MAG patients with an INCAT score > 3 (expression of clear motor impairment). Moreover, another possible explanation may be that autoantibodies are produced by plasma cells, which no longer express the CD20 antigen [58]. 

Recently, a retrospective analysis clarified the role of the reduction of anti-MAG IgM antibodies as a biomarker of response to immunotherapies: a sustained reduction of at least 50% compared with pretreatment levels could be considered a valuable indicator for the therapeutic response [59]. 

Interestingly, some patients, after rituximab infusion, can experience acute worsening associated with a strong increase in anti-MAG titers. This worsening is attributed to IgM flare [60,61,62,63] due to different mechanisms, such as B-lymphocyte lysis with resultant release of intracellular paraprotein, disruption of the idiotype/anti-idiotype network, or cytokine overproduction [61,64,65]. Fortunately, a successful treatment with plasma exchanges in patients presenting IgM flare has been reported, with outstanding and rapid neurological improvement [66].

Lastly, different treatment approaches are under investigation. Ibrutinib, an oral inhibitor of Bruton tyrosine kinase (Btk) used in patients affected by Waldenstrom macroglobulinemia associated with mutation in *MYD88* or *CXCR4* genes, demonstrated early improvement in three patients with anti-MAG neuropathy in [67]. A phase 1 study is ongoing to demonstrate the safety and efficacy of lenalidomide in a group of patients with anti-MAG demyelinating sensorimotor neuropathy (clinical trial identification number: NCT03701711). Recently, a novel approach based on the development of an antigen-specific molecule (namely, PPSGG) that serves as a decoy for an anti-MAG antibody through the recognition of the HNK1 epitope, demonstrated that it can significantly reduce >90% of a circulating anti-MAG antibody in the mouse [68].

## 3. Multifocal Motor Neuropathy with Conduction Block (MMNCB)

MMNCB is an acquired and rare neuropathy that typically involves the motor nerve fibers of the peripheral nervous system, sparing the sensory fibers [69,70].

It has an estimated prevalence of 0.6/100000 with onset in middle age (third to fifth decade, very rarely after 70 years old) and a clear prevalence in the male sex (male/female ratio: 3:1) [71].

### 3.1. Pathomechanism

MMNCB has been considered until recently as a “demyelinating” polyneuropathy due to the presence of conduction blocks (CBs). However, more in-depth knowledge of its pathomechanism lets neurologists be able to reclassify it into the new nosological entity of autoimmune nodopathy [1]. 

MMNCB is associated in over 70% with highly increased levels of specific antibodies anti-GM1. It is a ganglioside predominantly expressed in the motor axon membrane, and it is involved in the clustering of ion channels at the nodal/paranodal region. Therefore, the primary damage is not directed toward the myelin sheet, but the binding of IgM anti-GM1 antibodies causes mislocalization and internalization of sodium and potassium channels, preventing the transmission of the action potentials. Moreover, the second disease mechanism is represented by complement activation that mediates the formation of the membrane attack complex, compromising the membrane integrity and leading to axonal damage and loss [72,73,74]. These complex mechanisms result in myelin detachment in the nodal and paranodal regions, nodal lengthening, and disruption of ion channels determining altered membrane polarization and functional block of action potentials without real demyelination [75].

Nevertheless, around 30% are seronegative forms: in these cases, they have been found in the serum IgM anti-GM2 or anti-GD1b, which cross-reactive with GM1 ganglioside, or entirely seronegative forms, in which the trigger of the autoimmune reaction is not clear, but pathomechanism and clinical presentation are very similar to a seropositive form, with a predominant activation of a complement [74]. 

### 3.2. Clinical Features

MMNCB is a motor multifocal neuropathy characterized by a chronic course with asymmetric involvement of upper limbs at the onset. It is slowly progressive, very similar to CIDP with an even milder grade of disability, and it is typically worsened by cold [74]. Muscle weakness follows a multineuropathic distribution with a distal–proximal gradient; usually, muscles innervated by different terminal nerves but by the same root, trunk, or cord have a different grade of weakness. Muscles innervated by the same nerve but different roots have the same grade of weakness, localizing the primary damage on segments of peripheral nerves [72]. The onset typical affects upper limbs, very rare distal leg and never proximal leg muscles [73]. To achieve a definite diagnosis of MMNCB, it is necessary to demonstrate the involvement of at least two motor nerves without signs of sensitive dysfunction and upper motor neuron involvement. 

### 3.3. Electrophysiological Features

The hallmark of this syndrome is the presence of multiple motor conduction blocks (CBs) in nonentrapment sites with spared sensory conduction [72]. CBs are defined as a reduction of 50% of the proximal compound muscle action potential (CMAP) area with respect to the distal stimulation, with a distal CMAP amplitude of at least 1 mV [72]. A key feature that allows neurophysiologists to differentiate this syndrome from other demyelinating polyneuropathies is the absence of temporal dispersion [73], which relates to the pathological process of demyelination and remyelination. In fact, the CBs in MMNCB are due to not demyelination but a functional block of action potential transmission; this feature, together with the sparing of sensory conduction, stimulated across the sites of motor conduction blocks, represents a major criterion to differentiate MMNCB from CIDP (e.g., Lewis–Sumner syndrome).

However, some patients cannot manifest any detectable CBs for various reasons: CBs are in sites not routinely stimulated (very proximal, needing the combination of advanced neurophysiological techniques, such as triple stimulation, transcranial magnetic stimulation, or transcutaneous cervical root stimulation) or, conversely, are action-dependent blocks (not present at rest) in which only repeated proximal stimulation (or overmaximal exercise testing) determines a progressive decrease in CMAP amplitude until complete block [76]. Moreover, needle electromyography can show, over the reduction of recruitment at maximal activation, fibrillation and fasciculations (signs of axonal nerve damage) [75]. A normal CMAP in a weak muscle with neurogenic recruitment in EMG is very useful to differentiate MMNCB from motor neuron disease (MND), particularly when CBs are very proximal and not immediately identifiable with routine stimulation [77].

### 3.4. Biochemical Features

Routine labs do not show any hallmark of MMNCB, except a slight increase in muscle creatine kinase in two-thirds of patients or IgM monoclonal protein at immunofixation electrophoresis [72]; different from CIDP, the cerebrospinal fluid (CSF) protein level is elevated in only 30% of patients and did not reach a concentration over 1 g/l. The principal biochemical feature of MMNCB is the presence in most patients of serum IgM antibodies directed versus ganglioside GM1 proteins, typically revealed with good sensitivity by ELISA testing; less frequently in seronegative forms can be found antibodies directed against other membrane glycolipids [73,74]. Concerning anti-GM1 antibody testing, a significantly increased sensitivity has been reported when GM1 is combined with galactocerebroside (GalC) and may represent a preferred option for GM1 reactivity testing in MMNCB [78]. 

### 3.5. Radiological Features

MMNCB can be differentiated from CIDP by using contrast-enhanced MRI of the brachial plexus; MMNCB has a typical asymmetric root involvement with hyperintensities in T2-weighted sequences with contrast enhancement in the T1 series. Moreover, ultrasonography usually shows mainly proximal nerve enlargement in nonentrapment sites, and altered radiological findings correspond to clinically involved nerves [72,79].

### 3.6. Differential Diagnosis

MMNCB usually goes into differential diagnosis with both inflammatory neuropathies and MND variant lower motor neurons (LMNs). The clinical presentation could be confused with an atypical motor neuron disease (MND) (the presence of fasciculation, muscle cramps, wrist or foot drop, and finger weakness) or motor polyneuropathy (chronic course, relative preservation of muscular trophism, and absence of spastic hypertonia) [70]. 

Nevertheless, clinical presentation and NCS help us to differentiate and identify MMNCB: the presence of conduction blocks without a clear reduction of CMAP amplitude with normal muscular trophism with respect to the severity of weakness [73] and the absence of bulbar and/or upper motor neuron signs help physicians to differentiate MMNCB from MND. In the case of clinical LMN syndrome without demonstrable CBs at NCS with asymmetric involvement of upper limbs and a slowly progressive course, the IVIg cycle is worth trying, which helps the physician to differentiate MND from MMNCB, which is responsive to treatment and has a dramatically different prognosis. 

Conversely, the absence of temporal dispersion and sensory involvement is a strong feature against CIDP variants, as motor CIDP (typically symmetric and with sensory involvement at NCS) and Lewis–Sumner syndrome (clinical and neurophysiological sensory involvement with frequently proximal leg muscle involvement) [71,72]. 

Furthermore, is possible to differentiate MMNCB from CIDP by using contrast-enhanced MRI of the brachial plexus; in fact, MMNCB has a typical asymmetric root involvement, while CIDP shows a symmetric presentation. Moreover, the ultrasound evaluation exhibits a larger cross-sectional area of the nerve segment with CBs in CIDP patients with respect to those with MMNCB [79]. 

Another possible differential diagnosis can be HNPP, which can be characterized by muscle palsy and conduction blocks at NCS; nevertheless, in HNPP, palsy has an acute onset, CBs are in the site of entrapment, sensory involvement is predominant, and a familiar history is typically positive [70].

### 3.7. Therapy

The treatment of choice is IVIg, followed eventually by SCIg as maintenance therapy [80] with an improvement rate of around 80%. Steroids and plasmapheresis are not recommended and can worsen patients’ conditions; the second line of treatment consists of immunosuppressive drugs, such as cyclophosphamide, mycophenolate mofetil, azathioprine, anti-CD-20 monoclonal antibodies such as rituximab [73,81]. Based on pathomechanism, very recently, eculizumab, a monoclonal antibody binding C5 human complement factor, has been tested with controversial results [74,75].

## 4. Autoimmune Paranodopathies

Paranodopathies represent a novel pathogenetic diagnostic category, distinct from common inflammatory neuropathies. The identification of antibodies directed against target structures of the paranodal region has allowed for understanding the peculiar characteristics of this group of neuropathies that previously fell into the category of CIDP.

The cell adhesion molecules contactin 1 (CNTN1) and contactin-associated protein 1 (Caspr1) on the axonal side and neurofascin 155 (NF155) on the terminal myelin loops represent the essential proteins for the complex axoglial interactions configuring the nerve in three domains: nodes, paranodes, and internode [82]. 

The production of antibodies against these axoglial structures determines the disruption of the anatomy of the node of Ranvier altering the neurophysiology of nerve conduction by compromising the saltatory conduction of myelin fibers without inflammation.

### 4.1. Pathomechanism

Myelinated axons are organized into distinct domains characterized by specific molecular arrangements: nodes of Ranvier, paranodes, and internodes.

The node of Ranvier is a fundamental structure of the nervous system that provides rapid transmission of impulses through the genesis of the action potential [83]. At the node of Ranvier, myelin is disrupted, and the axolemma has the highest density of voltage-gated sodium channels (Nav). The nodes are limited by paranodal junctions composed of three major proteins: CNTN1 and Caspr1 on the axonal membrane and NF155 on the terminal myelin loops. The paranodal junction acts as a barrier by limiting the mobility of ion channels and membrane proteins between the node and the internode [84,85].

About 10 years ago, the pathogenic role of the antibodies anti-CNTN1, anti-Caspr1, and NF-155 was associated with 5–10% of CIDP patients [86,87]. It is noteworthy that these antibodies against axoglial proteins are predominantly IgG4 isotype, therefore, unable to activate a complement but can block critical functions of the target antigen through multiple mechanisms by which autoantibodies can disrupt cellular functioning by blocking protein–protein interactions and ion channels rather than binding to the immunoglobulin Fc receptor [88]. These antibodies cross the permeable blood–nerve barrier at the dorsal root ganglia, nerve roots, and endplate regions reaching the nodes [89]. 

The presence of these antibodies directed against paranodal axoglial proteins results in disassembling of the paranode architecture with loss of transverse bands, terminal myelin detachment with consequent nodal enlargement, and axonal loss without any morphological features of segmental demyelination and inflammatory cells as occurs in CIDP. 

The disruption of the nodal region increases the nodal capacity and reduces the capacitive current, which may reduce paranodal transverse resistance with increased current leakage, radial shunt, and current backflow to the paranode instead of longitudinal progression to the next node [90,91]. Moreover, voltage-gated K channels (Kv) are abnormally expressed at the paranode, compromising the genesis of the action potential at the nodal level and hyperpolarizing the membrane, increasing the time required to depolarize the next node. All these electrical changes result in slow conduction and failure to transmit the impulse. All these considerations explain the electrophysiological features that can determine “demyelinating” neurophysiological findings in the presence of only paranodal dismantling without evidence of inflammatory cells and real de-remyelinating features.

### 4.2. Clinical Features

Patients with autoimmune paranodopathies generally are young, which can manifest with an acute–subacute onset (30–40%) (Guillain–Barré syndrome (GBS)–like) [87,92], usually associated with an early axonal degeneration, severe course, great disability at onset, and poor or transient response to IVIg. Each antibody can lead to a distinct and peculiar phenotype. 

Anti-CNTN1 are the first paranodal antibodies associated with CIDP patients [86]. Patients display an acute–subacute onset (GBS-like) associated with prematurely axonal motor impairment, sensory ataxia, postural tremor, facial weakness, and respiratory failure. Up to 60% of patients may show concurrent membranous glomerulonephritis due to the deposition of the IgG4 antibody along the glomerular basement membrane [43,93,94]. Both neurological and nephrological conditions are not responsive to standard immunotherapy (e.g., IVIg or steroids) but shows long-term remission with rituximab. Studies in animal models support that IgG3 subclass anti-CNTN1 antibodies may mediate the acute phase of the disease and can predict a temporary response to IVIg during the acute stage [95].

Anti-Caspr1 antibodies are described in around 2% of CIDP and 4% of acute-onset CIDP patients. The anti-Caspr1 phenotype is characterized by an acute or subacute onset that may be misdiagnosed with GBS in 50% of cases. Clinical features include early axonal motor involvement, sensory ataxia, cranial nerve deficit, or respiratory difficulty, and patients can report pain by up to 50% [43,87,96]. It is noteworthy that patients with an anti-Caspr1 IgG3 subclass display an acute and monophasic course indistinguishable from GBS, who respond to IVIG [97]. A longitudinal study based on the identification of IgG subclasses in a patient with acute-onset anti-Caspr1 neuropathy demonstrated that IgG3 of the acute phase switched to IgG4 in the chronic stage, which may explain why patients in the acute phase respond better to the standard treatment with respect to the chronic phase of the disease [97].

Anti-NF155 antibodies are present in around 5–10% of CIDP patients [85,87,98]. With respect to the other form of autoimmune paranodopathies, anti-NF155 patients are young, with a chronic progressive course [43,99], and only about 10% of patients can be misdiagnosed with GBS at onset [99]. Anti-NF155 is characterized by predominantly distal motor impairment, severe ataxia, and intentional and postural tremor associated with cerebellar features [98,99]. About 25% of patients display cranial nerve involvement (e.g., facial weakness and ophthalmoparesis). Moreover, in [100,101], some patients were described with concomitant central demyelination and optic neuritis. 

### 4.3. Electrophysiological Features

NCS in patients with autoimmune paranodopathies shows the typical demyelinating features that can resemble those found in CIDP patients. In fact, the electrophysiology shows prolonged distal motor latency, slow conduction velocity, conduction block, and some temporal dispersion. However, although the electrophysiological findings point to demyelination, pathology shows only paranodal dismantling without evidence of segmental de-remyelination, and not even inflammatory, because of the peculiar features of IgG4 [102]. 

However, some peculiar electrophysiological features can help in raising the suspicion of autoimmune paranodopathies. First, lacking the de-remyelination phenomenon, temporal dispersion occurs more rarely in this condition rather than in patients with typical CIDP. When temporal dispersion is present, it may be explained by the differential involvement of paranodes among nerve fibers with increased dyssynchronization of conduction velocities [102]. Moreover, patients seropositive for a paranodal antibody exhibit a greater axonal loss with reduced amplitude of the CMAP and spontaneous activity (e.g., fibrillation potentials) on needle electromyography [103,104]. In fact, as occurs in other nodopathies (e.g., AMAN), also the autoimmune paranodopathies share the disease mechanism of a pathophysiological continuum from a transitory conduction block to an early axonal degeneration [82]. Lastly, the resolution after immunotherapies of a conduction block without the appearance of temporal dispersion indicates a dysfunction of the node of Ranvier rather than a demyelinating process. In this perspective, follow-up electrophysiological tests are required to distinguish the reversible conduction failure (RCF) from a classic demyelinating conduction block [105].

### 4.4. Biochemical Features

Laboratory work-up shows marked elevated CSF protein levels (>1–2 g/l) with respect to patients with CIDP. Patients with anti-CNTN1 can show high proteinuria levels as nephrotic syndrome is frequently associated. 

Antibodies to paranodal antigens are identified in serum and sometimes are also detectable in the CSF, indicating a severe disruption of the blood–nerve barrier [106]. As antibodies recognize their target epitopes in their native configuration, the most sensible techniques are cell-based assays (CBAs) [107] or teased-nerve immunohistochemistry. Additionally, ELISA with the human recombinant protein represents a good screening technique with high sensitivity and specificity. Since anti-Caspr1 testing has lower sensitivity, probably because the presence of CNTN1 is required for an appropriate expression of Caspr1 in the membrane surface [108], it is recommended to execute a Caspr1 assay (ELISA or CBA) with cells cotransfected with Caspr1/CNTN1 [96]. Anyway, it is highly recommended to repeat the detection of the antibodies with a second confirmatory assay to guarantee their specificity, since false positives have been described [1,109]. Moreover, the longitudinal analysis of antibody titers allows monitoring of treatment response and early recognition of relapses [110]. 

Once paranodal antibodies are confirmed, the subclass study and antibody titers also can provide clinically relevant information that can be useful in clinical practice, improving the diagnostic accuracy, predicting prognosis, and guiding treatment choice [87]. In fact, patients with the IgG3 subclass are associated with an acute onset and more critical disease but eligible for IVIg treatment. 

### 4.5. Radiological Features

To our knowledge, only few data are available concerning the imaging of paranodopathies. The MRI and ultrasound patterns seem to be similar to that seen in typical CIDP. Diffuse symmetric enlargement of lumbosacral plexus/roots with gadolinium enhancement is a common MRI finding in patients with paranodopathies (82% in anti-NF155, 50% in anti-CNTN1) [111], and nerve ultrasound showed multifocal and asymmetric swelling with hypoechogenic nerves, as expected in inflammatory neuropathies, in [112]. However, no sufficient longitudinal studies on the imaging of paranodopathies are present to draw conclusions. 

### 4.6. Differential Diagnosis

The most frequent differential diagnosis of autoimmune paranodopathies is represented by CIDP. The clinical phenotype and electrophysiological features should lead the correct diagnosis. CIDP guidelines (EAN/PNS 2021) recommend detecting paranodal antibodies in patients with clinical CIDP, resistant to standard immunomodulating therapies (IVIG or steroids), acute or subacute onset (GBS-like or acute-onset CIDP), low-frequency tremor or ataxia, predominantly distal motor impairment, respiratory deficit, and cranial nerve involvement, associated with nephrotic syndrome and high CSF protein levels [1]. Therefore, in the case of patients with clinical and electrophysiological features resembling CIDP but without response to immunotherapy (steroids, IVIg, or plasma exchange), we strongly suggest testing antiparanode antibodies. However, the test is not widely diffuse, and its use is mainly for academic purposes. For this reason, we suggest trying rituximab treatment for those patients not responsive to standard immunotherapy in order to avoid further axonal loss and disability burden. 

### 4.7. Therapy

Autoimmune paranodopathies rarely (20%) respond to IVIg, though about 50–60% may have a clinical response to steroids or plasma exchange [43,98,99]. IgG subclass identification can predict the immunotherapeutic response since patients with the IgG3 antibody can respond to IVIg treatment. 

Although randomized controlled trials have not been performed, case series reported a good response to rituximab in around 90% of patients, with complete remission in about 80%, and some case reports described clinical relapse during follow-up in [99,113]. Response to rituximab may be slower if there is severe axonal degeneration, and in some patients, the onset of improvement may be delayed by months [114]. Clinical response is related to reduction of serum neurofilament light chain and antibody titers [99,110]. Therefore, a longitudinal study of antibody titers should allow the monitoring of response of treatment but also is able to identify relapses early.

## 5. When and Why to Look for Antibody in Chronic Dysimmune Autoantibody-Related Neuropathy?

In conclusion, autoimmune neuropathies associated with antibodies must be considered an independent pathological entity that needs to be properly recognized as properly treated (Table 1).

An anti-MAG antibody should be tested, regardless of the detection of IgM monoclonal gammopathy, in patients complaining distal sensory symptoms, mild weakness, sensory ataxia, and hand tremor that show at NCS a demyelinating neuropathy more pronounced in distal segments (abnormal prolonged DML). A cut-off of >7000 BTU represents the best cut-off in recognizing anti-MAG neuropathy. The correct diagnosis of anti-MAG neuropathy is essential first to the treatment choice and to ensuring an accurate hematological follow-up in these patients as well.

An anti-GM1 antibody should be tested in patients with chronic focal motor deficit, typically involving the upper limb with an asymmetric pattern, without significant symptoms or signs of sensory impairment. The electrophysiological hallmark is represented by motor CBs with complete sparing of sensory nerve fibers. The correct diagnosis is essential for the treatment choice, as patients can worsen with steroids and plasmapheresis.

An anti-paranodal protein antibody (anti-CNTN1, anti-Caspr1, and anti-NF155) should be tested in patients complaining of sensorimotor impairment, often with acute/subacute onset, severe disability, ataxia, tremor, respiratory failure, and cranial nerve involvement, associated with nephrotic syndrome and very high CSF protein levels. Again, the correct diagnosis is essential for the treatment choice, as patients do not respond to IVIg and steroid, while high response frequency can be achieved with rituximab therapy.

## Figures and Tables

**Table 1 brainsci-13-00037-t001:** Main features of chronic dysimmune autoantibody-related neuropathies.

Differential Cues with Respect to CIDP	Anti-MAG Neuropathy	MMNCB	Paranodopathies
**Clinical**	Predominantly sensory polyneuropathy with sensory ataxia and tremor in the upper limbs	Motor multifocal neuropathy with asymmetric involvement of upper limbs	Sensorimotor neuropathy with possible acute/subacute onset, associated with tremor, ataxia, and distal weakness
**Electrophysiological**	Demyelinating neuropathy with disproportionate distal slowing (abnormal TLI) and slower CV in entrapment site	Multiple motor CBs in non-entrapment sites with spared sensory conduction study	Slowing of CV in the range of demyelination without temporal dispersion and greater axonal loss
**Laboratory**	M-protein	CSF protein can be increased (<1 g/l)	CSF protein >1–2 g/l Proteinuria (anti-CNTN1)
**Testing**	Anti-MAG (≥7.000 BTU)	Anti-GM1 (combined with GalC)	Anti-CNTN1 Anti-Caspr1 Anti-NF155
**Treatment**	Rituximab	1°: IVIg (SCIg) 2°: immunosuppressive drug (rituximab)	Rituximab

BTU = Buhlmann titer units; CB = conduction block; CIDP = chronic inflammatory demyelinating polyneuropathy; CSF = cerebrospinal fluid; CV = conduction velocity; GalC = galactocerebroside; IVIg = intravenous immunoglobulin; MMNCB = multifocal motor neuropathy with conduction block; SCIg = subcutaneous immunoglobulin; TLI= terminal latency index.

## Data Availability

Not applicable.

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
