# Peer review of "Value of Antibody Determinations in Chronic Dysimmune Neuropathies"

_brainsci, 2022, doi:10.3390/brainsci13010037_

Round 1
Reviewer 1 Report
Comments and Suggestions for Authors
Tozza et al, wrote a review on the clinical impact of antibodies in chronic dysimmune neuropathies.
This is a timely and original review on an interesting topic. The originality consists on an antibody-based approach to neuropathies. The review is updated, clear and well written. Nonetheless I have some observations:
1. The first sentence of the abstract and introduction: I would delete “obviously”
2. Anti-MAG neuropathy:
a. the statements that it is “slowly progressive over decades” is misleading since patients may complain symptoms onset or worsening over some months/few years.
b. Also, “marked distal atrophy” is not a clue clinical feature of this disease.
c. Functional impairment of the hands may also be listed among clinical features.
d. Pathological mechanism: MAG has already been abbreviated, no need to spell it again
e. Hematological conditions associated to anti-MAG neuropathy should be further listed and described.
f. The cut-off of 7000/10000 BTU should be clarified and discussed and the clinical implications of false positive and false negative should be discussed.
g. In the “Biochemical Features” “sensitive” is repeated twice (line 106 and 107, please delete the second one)
h. Therapy: The last Cochrane review should be quoted Lunn MP, Nobile-Orazio E. Immunotherapy for IgM anti-myelin-associated glycoprotein paraprotein-associated peripheral neuropathies. Cochrane Database Syst Rev. 2016 Oct 4;10(10):CD002827. doi: 10.1002/14651858.CD002827.pub4. PMID: 27701752; PMCID: PMC6457998.
i. The last section on novel treatment approaches should be clarified: Which are the mechanisms? are there preliminary clinical data? Which patients should be selected for new treatments? References are missing.
3. Multifocal motor neuropathy with conduction block
a. Authors should clarify if the definition “multifocal motor neuropathy with conduction block” is the same than “multifocal motor neuropathy” and should be consistent throughout the text (line 284 there is “MMN” instead of “MMNCB”).
b. Page 5, line 194: is it correct neuropathies? Or the authors want instead to write nodopathies?
c. References 63,65,70,71 appear incorrect. References should be carefully checked throughout the whole manuscript. Authors should refer, if related to a specific result, to the original study rather than to a review.
d. With regard to the anti-GM1 antibody testing, an increased sensitivity has been reported when the GM1 is combined with galactocerebroside. The following reference should be included: Nobile-Orazio E, Giannotta C, Musset L, Messina P, Léger JM. Sensitivity and predictive value of anti-GM1/galactocerebroside IgM antibodies in multifocal motor neuropathy. J Neurol Neurosurg Psychiatry. 2014 Jul;85(7):754-8. doi: 10.1136/jnnp-2013-305755. Epub 2013 Aug 1. PMID: 23907602.
4. Morphological features (nerve US and MRI) are largely neglected in this review. Authors may add a specific paragraph for each condition or include such information within the other sections.
5. Typing: to be checked (e.g. “ani-Caspr1” line 319) as well as grammar (first page, line 33: crucial for properly diagnosing; line 37: associated with)
Author Response
Tozza et al, wrote a review on the clinical impact of antibodies in chronic dysimmune neuropathies.
This is a timely and original review on an interesting topic. The originality consists on an antibody-based approach to neuropathies. The review is updated, clear and well written. Nonetheless I have some observations:
- The first sentence of the abstract and introduction: I would delete “obviously”
Response: we deleted it as suggested.
- Anti-MAG neuropathy:
- the statements that it is “slowly progressive over decades” is misleading since patients may complain symptoms onset or worsening over some months/few years.
- Also, “marked distal atrophy” is not a clue clinical feature of this disease.
- Functional impairment of the hands may also be listed among clinical features.
Response: we modified as reviewer suggested (lines 83-86).
- Pathological mechanism: MAG has already been abbreviated, no need to spell it again
Response: we abbreviated it as reviewer correctly suggested.
- Hematological conditions associated to anti-MAG neuropathy should be further listed and described.
Response: we further listed the hematological conditions (lines 161-162).
- The cut-off of 7000/10000 BTU should be clarified and discussed and the clinical implications of false positive and false negative should be discussed.
Response: we discussed the issue of cut-off (7.000 vs 10.000 BTU) and the implications of false positive/negative (lines 112-120).
- In the “Biochemical Features” “sensitive” is repeated twice (line 106 and 107, please delete the second one)
Response: we deleted the repeated “sensitive” according to reviewer’s suggestion.
- Therapy: The last Cochrane review should be quotedLunn MP, Nobile-Orazio E. Immunotherapy for IgM anti-myelin-associated glycoprotein paraprotein-associated peripheral neuropathies. Cochrane Database Syst Rev. 2016 Oct 4;10(10):CD002827. doi: 10.1002/14651858.CD002827.pub4. PMID: 27701752; PMCID: PMC6457998.
Response: we cited (reference # 56) the last Cochrane review according to reviewer’s suggestion.
- The last section on novel treatment approaches should be clarified: Which are the mechanisms? are there preliminary clinical data? Which patients should be selected for new treatments? References are missing.
Response: we briefly described the preliminary data currently available about novel treatment approaches as reviewer requested (lines 204-213).
- Multifocal motor neuropathy with conduction block
- Authors should clarify if the definition “multifocal motor neuropathy with conduction block” is the same than “multifocal motor neuropathy” and should be consistent throughout the text (line 284 there is “MMN” instead of “MMNCB”).
Response: we corrected the “MMN” in “MMNCB”.
- Page 5, line 194: is it correct neuropathies? Or the authors want instead to write nodopathies?
Response: we specified that MMNCB should be considered as “nodopathy” (line 225).
- References 63,65,70,71 appear incorrect. References should be carefully checked throughout the whole manuscript. Authors should refer, if related to a specific result, to the original study rather than to a review.
Response: we corrected the references and checked them throughout the whole manuscript.
- With regard to the anti-GM1 antibody testing, an increased sensitivity has been reported when the GM1 is combined with galactocerebroside. The following reference should be included: Nobile-Orazio E, Giannotta C, Musset L, Messina P, Léger JM. Sensitivity and predictive value of anti-GM1/galactocerebroside IgM antibodies in multifocal motor neuropathy. J Neurol Neurosurg Psychiatry. 2014 Jul;85(7):754-8. doi: 10.1136/jnnp-2013-305755. Epub 2013 Aug 1. PMID: 23907602.
Response: we added this concept (lines 286-289) and the reference (reference #78) as well.
- Morphological features (nerve US and MRI) are largely neglected in this review. Authors may add a specific paragraph for each condition or include such information within the other sections.
Response: we added for each chapter the paragraph “radiological features” (lines 134-143; 290-296; 455-463).
- Typing: to be checked (e.g. “ani-Caspr1” line 319) as well as grammar (first page, line 33: crucial for properly diagnosing; line 37: associated with)
Response: we corrected the typos.

Reviewer 2 Report
Comments and Suggestions for Authors
This is an interesting and informative review. However, the title may be a bit misleading. Nosology and nomenclature should follow that used by the EAN/PNS guidelines ...on diagnosis and treatment of CIDP, 2nd Revision (Van den Bergh et al 2021, cited by the authors as ref #1). A few additions would make this review more valuable and useful to clinicians.
1. In particular, the term "Chronic Dysimmune Neuropathies" does not have an official definition, and would be taken by many clinicians and investigators to include CIDP. A better title and approach might be: "Value of Antibody Determinations in Chronic Dysimmune Neuropathies other than CIDP: Anti-MAG Neuropathy, Motor Neuropathy with Conduction Blocks and Autoimmune Paranodal Neuropathies". These are nicely and concisely distinguished and summarized in Section 5 of the current manuscript.
2. It would be most helpful if the authors included a table clearly laying out for each of these three entities:
a. Differences from typical CIDP, with clinical and laboratory
findings concisely listed
b. Antibody (ies) to seek and preferred test(s)
c. Recommended first line treatment
3. EAN/PNS guidelines suggest that "Monoclonal gammopathy testing should be performed in all patients with suspected CIDP" (Author's ref 1). " This should be explicitly repeated in this review, as should the additional specification that "if an IgM paraprotein is present, anti-MAG antibodies should be tested".
4. There should be some discussion of how and when to proceed if the patient is presenting with a condition resembling CIDP but fails to respond to recommended treatment for that condition (for example- section 4.5, lines 422-424).
5. Since anti-CD20 antibodies and rituximab is mentioned in several places, the authors should briefly discuss the possibilities that failure to respond to rituximab may be due to production of the autoantibodies by plasma cells, which no longer express CD20 (see MJ Leandro: Arthritis Research and Therapy 2013, Vol 15, Suppl 3).
6. In the case of paranodopathies, the authors should discuss the multiple mechanisms by which autoantibodies can disrupt cellular functioning other than complement activation or Fc receptor-mediated inflammation, for example by blocking protein-protein interactions and ion channels. A reference for their assertion that IgG4 can act through "binding to the immunoglobulin Fc receptor" should be added and critically evaluated, or softened. This is particularly important since Fc Receptor antagonists are now coming into clinical use but may be ineffective if binding of the autoantibody alone is sufficient to inhibit or alter the target's function.
Author Response
Reviewer #2
This is an interesting and informative review. However, the title may be a bit misleading. Nosology and nomenclature should follow that used by the EAN/PNS guidelines ...on diagnosis and treatment of CIDP, 2nd Revision (Van den Bergh et al 2021, cited by the authors as ref #1). A few additions would make this review more valuable and useful to clinicians.
- In particular, the term "Chronic Dysimmune Neuropathies" does not have an official definition, and would be taken by many clinicians and investigators to include CIDP. A better title and approach might be: "Value of Antibody Determinations in Chronic Dysimmune Neuropathies other than CIDP: Anti-MAG Neuropathy, Motor Neuropathy with Conduction Blocks and Autoimmune Paranodal Neuropathies". These are nicely and concisely distinguished and summarized in Section 5 of the current manuscript.
Response: we thank the reviewer for her/his comment. We changed the title as “Value of Antibody Determinations in Chronic Dysimmune Neuropathies”.
- It would be most helpful if the authors included a table clearly laying out for each of these three entities:
- Differences from typical CIDP, with clinical and laboratory findings concisely listed
- Antibody (ies) to seek and preferred test(s)
- Recommended first line treatment
Response: we added a summarized table with the main features of each chronic dysimmune autoantibody-related neuropathy (table 1)
- EAN/PNS guidelines suggest that "Monoclonal gammopathy testing should be performed in all patients with suspected CIDP" (Author's ref 1). " This should be explicitly repeated in this review, as should the additional specification that "if an IgM paraprotein is present, anti-MAG antibodies should be tested".
Response: we added what reviewer correctly highlighted (lines 55-57).
- There should be some discussion of how and when to proceed if the patient is presenting with a condition resembling CIDP but fails to respond to recommended treatment for that condition (for example- section 4.5, lines 422-424).
Response: we discussed how and when to proceed if the patient is presenting with a condition resembling CIDP but fails to respond to treatment (lines 472-477).
- Since anti-CD20 antibodies and rituximab is mentioned in several places, the authors should briefly discuss the possibilities that failure to respond to rituximab may be due to production of the autoantibodies by plasma cells, which no longer express CD20 (see MJ Leandro: Arthritis Research and Therapy 2013, Vol 15, Suppl 3).
Response: we thank the reviewer for her/his comment. We discussed possibilities that failure to respond to rituximab may be due to production of the autoantibodies by plasma cells, which no longer express CD20 (lines 191-192). We added the suggested reference as well (reference #58).
- In the case of paranodopathies, the authors should discuss the multiple mechanisms by which autoantibodies can disrupt cellular functioning other than complement activation or Fc receptor-mediated inflammation, for example by blocking protein-protein interactions and ion channels. A reference for their assertion that IgG4 can act through "binding to the immunoglobulin Fc receptor" should be added and critically evaluated, or softened. This is particularly important since Fc Receptor antagonists are now coming into clinical use but may be ineffective if binding of the autoantibody alone is sufficient to inhibit or alter the target's function.
Response: we added different mechanisms in paranodopathies and we softened the role of FcR in the disrupting cellular functioning (lines 357-360).
